# Antibody-Based Immunotherapies as a Tool for Tackling Multidrug-Resistant Bacterial Infections

**DOI:** 10.3390/vaccines10111789

**Published:** 2022-10-25

**Authors:** António M. M. Seixas, Sílvia A. Sousa, Jorge H. Leitão

**Affiliations:** 1Department of Bioengineering, IBB—Institute for Bioengineering and Biosciences, Instituto Superior Técnico, Universidade de Lisboa, Av. Rovisco Pais, 1049-001 Lisboa, Portugal; 2Associate Laboratory, i4HB—Institute for Health and Bioeconomy, Instituto Superior Técnico, Universidade de Lisboa, Av. Rovisco Pais, 1049-001 Lisboa, Portugal

**Keywords:** antibody-based immunotherapies, multidrug-resistant (MDR) bacterial infections, monoclonal antibodies (mAbs), immunoglobulin Y (IgY), nanobodies, polyclonal antibodies (pAbs)

## Abstract

The discovery of antimicrobials is an outstanding achievement of mankind that led to the development of modern medicine. However, increasing antimicrobial resistance observed worldwide is rendering commercially available antimicrobials ineffective. This problem results from the bacterial ability to adapt to selective pressure, leading to the development or acquisition of multiple types of resistance mechanisms that can severely affect the efficacy of antimicrobials. The misuse, over-prescription, and poor treatment adherence by patients are factors strongly aggravating this issue, with an epidemic of infections untreatable by first-line therapies occurring over decades. Alternatives are required to tackle this problem, and immunotherapies are emerging as pathogen-specific and nonresistance-generating alternatives to antimicrobials. In this work, four types of antibody formats and their potential for the development of antibody-based immunotherapies against bacteria are discussed. These antibody isotypes include conventional mammalian polyclonal antibodies that are used for the neutralization of toxins; conventional mammalian monoclonal antibodies that currently have 100 IgG mAbs approved for therapeutic use; immunoglobulin Y found in birds and an excellent source of high-quality polyclonal antibodies able to be purified noninvasively from egg yolks; and single domain antibodies (also known as nanobodies), a recently discovered antibody format (found in camelids and nurse sharks) that allows for a low-cost synthesis in microbial systems, access to hidden or hard-to-reach epitopes, and exhibits a high modularity for the development of complex structures.

## 1. Introduction

The use of antibodies is considered one of the first consistently effective antimicrobial strategies developed. In the 1890s, specific antibodies were found to be able to protect against bacterial toxins, which in turn led to the development of antibody treatments for diverse infectious diseases [1]. The initial antibody preparations used were derived from the serum of immune human donors or immunized animals. This approach, known as serum therapy, was somewhat effective but due to the extensive quantities of exogenous proteins administrated to the patients, it was also highly prone to side effects such as hypersensitivity reactions and serum sickness, a form of antigen–antibody complex disease [2]. Improvements to these techniques were consistently being achieved, reducing some of the side effects. However, a rapid decline of this approach occurred with the availability of the first antimicrobials, which completely changed medicine forever and represents one of the most extraordinary accomplishments in this field [2,3]. The use of serum-based approaches for antibacterial treatments was predominantly discarded, retaining a small slot as a treatment for venoms and toxins.

The rise of antimicrobials was an incredible feat of mankind that led to the development of modern medicine, an increase in life expectancy, with an important economic impact in terms of hospital and treatment costs [4]. The employment of antimicrobials is done all throughout modern medicine and has saved numerous lives. However, this great success has led to the development of a new major threat to global public health, known as antimicrobial resistance, which is rendering many commercially available antimicrobials ineffective [5]. This issue stems from the bacteria’s remarkable ability to adapt to the selective pressure presented by antimicrobials.

Bacteria can develop or acquire multiple types of resistance mechanisms that can severely hinder the efficacy of the antibiotic. This issue is being exacerbated by their misuse, over-prescription, and poor treatment adherence by patients [3,6]. The antimicrobial-resistant problem has been increasing over decades, leading to the development of an epidemic of infections untreatable by first-line therapies. This crisis is especially centered on the multidrug-resistant “ESKAPE” bacteria (*Enterococcus* spp., *Staphylococcus aureus*, *Klebsiella* spp., *Acinetobacter baumannii*, *Pseudomonas aeruginosa* and *Enterobacter* spp.) [7]. In some cases, such as that of *A. baumannii*, strains resistant to all available antibiotics are already being isolated [8]. 

Despite the increasing need for effective antimicrobials with novel modes of action, this development has been extremely scarce with very few new compounds being approved in the past 30 years. In fact, nearly all antibiotics or their derivatives currently being used in clinical situations were discovered in the decades of 1940 to 1960 [3,9]. Obviously, other alternatives are required to quench this increasing problem. Vaccination is an exceptionally effective approach and is often seen as the possible answer to the problem; however, its application to some of the most problematic multidrug-resistant bacteria has been unsuccessful, and methods of active immunization work better as prevention approaches, not having an immediate effect on the infection, which is required in a clinical situation. Opposite to active immunization, passive immunization offers an immediate protection against the infectious agent, although short-lived and lasting from several weeks to months [10]. 

## 2. Antibody-Based Immunotherapies—An Overview

The use of passive immunization approaches has many of the characteristics required to tackle multidrug-resistant bacteria, including specific activity, ability to target resistant bacteria, and unlikeliness to lead to the development of new antimicrobial resistance. Passive immunization can be simply described as the transfer of already formed antibodies to a receptor individual. Antibodies together with B and T cells constitute one of the most important components of the adaptive immune system. Antibodies or immunoglobulins (Igs) are Y-shaped glycoproteins capable of specifically recognizing antigens, attaching and forming a molecular link in the communication network of the remaining elements of the immune system, activating the complement components of the immune response, and leading to neutralization or elimination of foreign threats [11]. 

In mammals, immunoglobulin G (IgG) is the most common isotype. IgGs presents a tetrameric structure composed of two identical heavy chains and two identical light chains. The heavy chains are composed of four domains, three constant and one variable. The light chain has two domains, one constant and one variable. The properties that allow IgGs to recognize and bind to antigens are concentrated in short segments within the variable domains that, as the name indicates, have a high degree of variability [12] (Figure 1). Immunoglobulin M (IgM) is a different antibody isotype in mammals that presents a predominantly pentameric structure, with 10 or 12 antigen-binding sites [13]. Following exposure to foreign antigens, they are the first antibodies secreted. Typically, IgM show a lower antigen-binding affinity when compared with IgG. However, the polyvalent nature of their structure contributes for a high avidity binding and an efficient engagement of complement-dependent cell lysis. This higher avidity allows for a greater efficiency on the binding to antigens present at lower concentration, and to non-protein antigens such as lipids and carbohydrates [13]. 

Another antibody isotype currently being studied due to its specific characteristics is the immunoglobulin A (IgA). IgA has evolved to be secreted and function in mucosal surfaces, making them vital for protection against mucosal pathogens but also for the function of the healthy microbiome. The production and secretion of IgA is highly efficient, with plasma cells producing around 3 to 5 g each day, more than all other isotypes combined, with over one gram being secreted per day. Structurally its composed of four immunoglobulin domains and can be found in monomers, dimers, and as secretory. Dimerization and secretion is aided by the covalent attachment to the J chain, a critical step in transcytosis as it presents a bridge between the antibody and the polymeric immunoglobulin receptor (PIGR). Both IgA and IgM bind to PIGR; however, IgM binds transiently and IgA uses disulfide bonds to be covalently linked, this causes IgA not to be released from PIGR binding upon arrival at mucosal surfaces. PIGR is cleaved by an unknow protease but remains bound to IgA as a secretory factor and creates the secretory IgA (sIgA). sIgA is believed to have increased resistance to proteolytic degradation in the gastrointestinal tract, an integral characteristic given the high concentration of proteases in the lumen. This leads IgA to be more abundant at mucosal surfaces as IgM is more susceptible to proteolysis. sIgAs are considered the first barrier against pathogens in mucosal surfaces, despite lacking effector functions they can cause bacterial agglutination, disturb motility, neutralize toxins and inhibit adherence to epithelium, preventing circulatory dissemination. These potent antipathogen effects have been described against respiratory viruses and multiple gastrointestinal pathogens such as *Clostridioides difficile, Salmonella enterica* subsp. *enterica* serovar Typhimurium, and *Shigella flexneri.* [14,15]. 

The therapeutic benefits offered by antibodies result from an assortment of mechanisms. Antibodies can bind to antigens in bacteria preventing adherence and other important steps of infection or block receptor-ligand formation of toxins and viruses [16]. They can also use a different mechanism that is dependent on mediators, where the binding of the antibody to surface receptors leads to agglutination and the immobilization of the target cell. After this, an immune-complex with the antigen is formed, and initiation of antibody-mediated cellular cytotoxicity, complement-dependent cytotoxicity, or opsonization can occur [16,17,18] (Figure 1). All these different mechanisms of action cannot be obtained from a single antibody formulation. Monoclonal antibodies, for example, due to their specificity to a single epitope, are generally used to block receptor-toxin ligand formation but can also be used to treat cancer by targeting specific antigens on cancer cells, initiating antibody-mediated cellular cytotoxicity or complement-dependent cytotoxicity [16]. For the application of antibody therapies in infectious diseases, the suggested mechanisms are blockage of the pathogenic virulence mechanisms, such as the secretion of virulence factors, recruitment of immune mediators and effectors to eliminate the pathogen through phagocytosis, or usage of an established anti-cancer therapy where the antibody directly kills the pathogen by targeted delivery of radionuclides or toxic drugs [19]. 

Despite the increasing emergence caused by the worldwide growth of antimicrobial resistance, the relative inefficacy in immunocompromised patients of antimicrobial drugs and the many technological advances in the field of immunoglobulin research, the number of antibodies licensed for clinical use is scarce. Furthermore, the vast majority of these antibodies were developed for the treatment of non-infectious diseases, such as oncology, rheumatology, and transplant medicine [20]. The use of these approaches for bacterial infections should, in theory, be less problematic than those developed for non-infectious diseases. This assumption results from the fact that the antigens targeted by the antibody are extremely different from those found in the host, opposite to what happens in tumor treatment, where a discrimination between self-antigens is required [2,16]. 

Despite the low number of approved antibody-based immunotherapies for infectious diseases, there is evidence that antibodies can exert a protective effect against a variety of microorganisms, including intracellular ones [2,21]. Various studies show positive effects of using monoclonal or polyclonal antibodies to treat bacterial, viral, and fungal infections [19]. Another interesting approach is the use of a combination of both antibody-based immunotherapy and antimicrobials, which can result in a synergetic or additive effect, having a success rate higher than the separate use [22]. This opens the door to an easier incorporation of immunotherapies into the existing clinical protocols, but also shorter stays in intensive care units and reductions of morbidity, mortality, and health care costs. 

The characteristics of antibody-based therapies allow them to have some advantages over the typical use of antimicrobials. Antibodies are highly specific, allowing them to target only the microorganism intended, with minimal influence on other microorganisms present in the normal flora, and do not exert a selecting pressure for resistance on the non-targeted flora [6]. The use of these approaches has effects on the enhancement of bacterial clearance, prevention of colonization and invasion, and reduction of damage caused by cytotoxic or hyperinflammatory factors. Negative effects of existing antimicrobial resistances or the development of new resistances against these treatments are highly unlikely. 

The possible use of a combination of both antimicrobials and antibodies can result in a reduction of antimicrobials use and increase in their effectiveness, which may result in a lower selective pressure and decrease the prevalence of antimicrobials resistance [22,23]. The response of bacteria against antibodies is more tamed, resulting in less bacterial SOS responses often responsible for several side effects that may include toxin release and increased transfer of resistant genes [6,24]. Despite some exceptions, such as some nanobodies, antibodies are significantly more stable than antimicrobials, having half-lives from weeks to months, exceeding those of the antimicrobials that only maintain their activity in the patient for a few days [22,25,26]. Despite the advantages presented, some important limitations are associated with the antibody therapies. As natural products, the production must occur in cell lines or other live expression systems, requiring a purification step, raising concerns about possible contamination of the preparations by prions, viruses, or other infectious agents. Due to the high specificity mentioned previously, the usage of these approaches requires knowledge of the causative agent of infection, meaning that a fast microbiological diagnosis is always required before the initialization of therapy, especially because the highest efficiency of treatment is obtained for infections at the early stages. Fortunately, the development of PCR and other rapid diagnostic techniques provide fast platforms, helping the establishment of antibody-based therapies [2,4]. Indeed, these fast diagnostic methods allow the specific identification of the pathogen in a short time, enabling the accurate selection of a specific antibody-based therapy. Altogether, this combination increases the confidence and reliability of antibody-based therapies, thus contributing to their establishment. The specificity might also be problematic in the case of mixed infections or the case of pathogens with high antigenic variation. The most obvious solution to this question is the use of a cocktail of antibodies able to target different microorganisms or different antigens [27]. Logistically, these approaches might present some difficulties, namely as the antibodies are proteins, the administration and maintenance must be done in a medical facility, as administration is typically given intravenously, which is unpractical for non-hospitalized patients. However, recently, an increase in the delivery of antibodies through the subcutaneous and intramuscular routes has been registered. In addition, oral immunotherapy with antibodies is a topic being studied using different isotypes and sources of antibodies, and direct delivery with an aerosol to the infected airways has been shown to be effective [28,29,30,31]. 

The major drawback of antibody-based immunotherapy is the high cost of production, storage, and administration. The research required to find the best antigen target and the best binding spot for the antibody is also very labor intensive, time consuming, and costly. The scale-up required for the large-scale production is also extremely difficult and financially demanding, despite some improvements in culturing and processing increasing the yields obtained [32]. These higher costs of production combined with the high specificity indicate that a potential market for any given antibody will be considerably small, meaning that the development of these strategies will be significantly harder for infectious diseases that are not common enough to provide a financial reward. However, the development of an antibody for passive immunotherapy requires a noticeably shorter time and lower cost than that needed to develop a vaccine [2,33]. 

For immunotherapy purposes, there are different antibodies formats currently being studied for the development of new therapies to deal with bacterial infections (Figure 2). In this review, the potential of four types of antibody formats as alternatives to antimicrobial therapies is explored.

## 3. Conventional Mammalian Polyclonal Antibodies

The first antibody preparations used for passive immunization contained polyclonal antibodies derived from the sera of immunized animals or humans, and in some cases, convalescing patients [2]. Nowadays, pAbs are produced by injecting an immunogen into an animal to elicit an immune response against a specific antigen. After immunization, the polyclonal antibodies are purified to obtain a solution that is free from other serum proteins. These antibodies can be purified directly from an animal serum after immunization with the antigen of interest or the infectious agent. The term polyclonal is derived from the fact that these molecules are derived from multiple B cell clones and can recognize different epitopes from the same antigen. This occurs because different lymphocyte clones responding against different epitopes of the antigen are formed during the antibody response [11,34]. 

pAbs preparations are comprised of numerous antibodies differing in primary structure, isotype, specificity, and glycosylation of the constant region, an important factor for interaction with Fc receptors [35]. The protective effect of pAbs is observed naturally in the passive transference of maternal immunity to protect newborns during the more vulnerable phases of early life [36]. The preparation of pAbs for clinical uses starts with the immunization of an animal. As higher volumes of blood can be retrieved from larger animals, the traditional sources of these antibodies are horses, sheep, goats, and rabbits, rather than mice [34]. This production is relatively inexpensive, and the total amount produced is limited by the amount of blood removed from the animal during its lifetime. Once an animal is exhausted, new immunization in a different animal needs to be performed, which will not be identical to the previous, due to the stochastic nature of the adaptive immune response and the polyclonal nature of these antibody preparations [34]. The method through which pAbs are produced makes them cost-effective, able to be supplied in large quantities, and the usage of larger animals reduces the need for multiple batches to be validated. Presently, pAbs are a low-cost alternative to monoclonal antibodies which are usually used for snake bites and post-exposure prophylaxis of infectious diseases such as rabies, botulism, and diphtheria [37]. 

The ability of these antibodies to recognize several epitopes also challenges the pathogen capability to avoid antibodies by mutating the antigen [38]. The variability inherently present in pAbs preparations increases the biological effector functions, as multiples sub-isotypes of IgG antibodies are present (IgG1, IgG2, IgG3 and IgG4) providing in many cases added benefits and protective advantage [39]. The use of these pAbs is also associated with some limitations, namely the lack of standardization due to the lot-to-lot variation could lead to deficient efficacy and low content of specific antibodies. 

The adoption of approaches using pAbs has been greater for tackling virus such as influenza, HIV, MERS, and SARS, with some levels of success [19,35,36,39,40] but also bacterial toxins such as shiga toxin and toxins from Clostridium difficile [11,19]. The use for treatment of other bacterial infections has been less common. Our group recently showed that a pAbs against an OmpA-like protein from Burkholderia cenocepacia greatly impairs the ability of these bacteria to adhere and invade human epithelial cells in vitro [41]. Both the adhesion and invasion are highly important steps in the infection process.

The fact that the antibodies produced, usually, are non-human may lead to exacerbated immune responses in the patients. This last hurdle could be overcome with the production of humanized antibodies in the animals. Beigel et al. used a novel approach with transchromosomic cattle carrying a human artificial chromosome comprising the entire human immunoglobulin gene repertoire, human immunoglobulin heavy chain (IGH) locus from chromosome 14, and human k light chain (IGK) locus from chromosome 2.6 [40]. These authors reported the production of a highly potent and specific fully human polyclonal IgG using this system [40]. 

An example is the human polyclonal immune globulin preparation (Altastaph) that targets capsular polysaccharides of *S. aureus* and is being developed for *S. aureus* infections complicated by bacteremia. A phase II double-blind, placebo-controlled trial found that when compared to the placebo the patients had a shorter median time to the resolution of fever and a shorter length of hospital stay. Unfortunately, the study was not powered to show efficacy. Nonetheless, the preliminary findings and safety profile indicate the potential of this therapy as an adjunct to antimicrobials [42]. A different human polyclonal preparation named Veronate^®^ against *S. aureus* derived from donors with high titers of antibodies against a surface adhesin of *S. aureus* and Staphylococcus epidermidis also went into clinical trials. The phase II trial showed promising results for the highest dose used. However, a phase III trial (NCT00113191) evaluating the prevention of late-onset sepsis in very low birth weight infants showed no effect of the antibody treatment in reduction of *S. aureus* [43]. A possible explanation for the disappointing result is the requirement of a humoral, cellular, and phagocytic response for control of these infections, which is not properly provided by the pre-mature infants [19]. 

## 4. Conventional Mammalian Monoclonal Antibodies

The technology for monoclonal antibodies (mAbs) production was introduced in 1975 by Kohler and Milstein [44]. mABs were immediately recognized as powerful tools for the targeted treatment of various diseases. mAbs can be from different isotypes of antibodies from human or animal IgG, IgM, IgA, avian IgY or single domain antibodies. Their main difference from polyclonal is they are homogenous immunoglobulins that only recognize one epitope. Due to this feature, they have a higher specific activity than polyclonal preparations. In 1986, a decade after the introduction of the technology, the first mAb was approved for human use by the FDA (Muromonab-CD3) [45]. As the first molecules were produced using mouse cell lines, they would be recognized as foreign molecules by the host immune response leading to mild to harsh immune reactions. The response to this problem was the development of several platform technologies able to produce mouse chimeric, humanized, and human antibodies [22,45]. 

Chimeric antibodies are a combination of the murine variable domains fused to the human constant domains. These antibodies are produced by cloning the murine variable region heavy and light chains genes, amplified from the hybridoma, together with the constant region genes of human heavy and light chains into a plasmid. This plasmid is then transfected into bacteria where the antibodies are produced as inclusion bodies, and then purified for in vivo use. These molecules are around 70% human and possess a human Fc sequence, reducing the possibility of immunogenicity [46]. Humanized antibodies contain 85 to 90% of human sequences. Their production is achieved by replacing all the rodent sequences except the complementarity determining regions (CDRs) for human sequences. However, this process of humanization is technically challenging, the insertion of CDRs in a generic framework is insufficient, as antibody affinity sometimes relies on framework regions. This process leads to antibody activity losses. Conservation of a few rodent sequences in the framework is required to restore the binding [46,47]. The production of fully human monoclonal antibodies was achieved with the development of phage display platforms, and also with the advancements in transgenic mouse platforms [46]. mAbs are the most successful antibodies, being approved and used as a therapy for cancer, autoimmune disorders, cholesterol, and infectious diseases, with the majority of these approved mAbs being chimeric or humanized [46,47,48]. 

Currently, about 100 IgG mAbs are approved by the Food and Drug Administration (FDA) for therapeutic uses [20]. New techniques using transgenic animals for the isolation of fully human mAbs are consistently being optimized [48]. Formulations of monoclonal antibodies are superior to polyclonal ones, in terms of specific activity, safety, constancy, and homogeneity. mAbs are also highly stable, especially the immunoglobulin G1 isotype that can reach half-lives of up to 21 days [49]. However, homogeneity can sometimes be disadvantageous, especially against more complex bacteria, where the targeting of a single epitope is not enough, in terms of expression and conservation, to develop a proper therapeutic response [4,26]. In these cases, the use of multivalent formulations may be required. The use of several mAbs targeting different antigens leads to an even higher cost of treatment. This high cost of treatment is considered the major drawback of mAbs, increasing concerns about the viability for widespread adoption. However, due to the emergency of multidrug-resistance reported for several bacterial pathogens, the use of mAbs therapies as alternatives to antimicrobials is being extensively studied. 

Thus far, three mAbs licensed by the FDA against bacterial exotoxins have been approved for treatment and prophylaxis. Raxibacumab [50] and obiltoxaximab [51] are monoclonal antibodies that target the lethal toxin of Bacillus anthracis and were approved for treatment of anthrax inhalation. The third mAb approved is bexlotoxumab [52] and targets the enterotoxin B of *Clostridioides difficile*. Anti-toxin mAbs inhibit virulence, limiting damage to the host without causing a selective pressure. However, their ability to directly tackle acute diseases is limited, and bexlotoxumab is an example of this, as the therapy is not approved for treatment of initial infection or protection against infection. Instead, it is used for reduction of recurrence of infection, a clinical situation that is frequently observed in *C. difficile* infections. For this reason, much work has been focusing on antibodies against outer membrane proteins of bacteria, involved in adhesion, evasion of immune system, and other bacterial processes [53,54,55,56]. These proteins have important functions in the bacterium, acting not only as easy targets but also as effective ones likely conserved in different clinical strains [57]. 

*P. aeruginosa* has been one of the most studied organisms for the development of immunotherapies [58]. One of the most successful examples is the development of a mAb targeting the conserved PcrV protein of this bacterium. This protein is part of the type III secretion system, an important virulence factor aiding the delivery of exotoxins into the target cells. This antibody displays protection in several animal models [59] and the mode of action involves the neutralization of the T3SS secretion system function. Another mAb, a PEGylated Fab named KB001, went into a randomized, double-blind, placebo-controlled clinical trial and was shown to prevent ventilator-associated pneumonia caused by *P. aeruginosa* infections [60]. Unfortunately, the efficacy was low in patients suffering from cystic fibrosis (CF). This low efficiency might be related to the low levels of the T3SS protein found in the sputum of CF patients chronically infected [53]. The targeting of polysaccharides, such as lipopolysaccharide (LPS) and capsular polysaccharide (CPS), have also attracted the interest of many researchers since the beginning of immunotherapies. These polysaccharides are essential for many bacteria to avoid immune systems, increasing their potential for the development of immunotherapies, with antibodies attaching to the CPS, improving the opsonophagocytosis of evasive bacteria [57]. IgM antibodies may have ideal characteristics for targeting these molecules and are thought to have a higher cross-reactivity due to the lack of affinity maturation [61]. These isotypes are multimeric, combining multiple low affinity interactions to accomplish a high functional avidity. This makes IgM larger molecules with higher side effects and shorter half-lives, decreasing their desirability for immunotherapies. Furthermore, the targeting of a polysaccharide antigen might lead to a shift in bacterial population from the targeted polysaccharide, as observed in *Streptococcus pneumoniae* strains in response to vaccination [62]. An example of a polysaccharide target against *P. aeruginosa* used an IgM mAb, denominated Panobacumab (KBPA-101), targeting the O-antigen of serotype O11. Panobacumab mediates protection in several murine models, has bactericidal and opsonophagocytic activities, and was shown to be safe in healthy volunteers [63]. It later underwent a phase IIa trial with 18 patients with nosocomial pneumonia caused by *P. aeruginosa* serotype O11. The antibody was safe and well tolerated, with 100% survival, where a 31% mortality was predicted on APACHE II scores. After 9 days, the pneumonia was considered resolved, compared to the standard-of-care of around 15 days [64,65]. 

Another bacterium also highly studied for the development of immunotherapies is *S. aureus*. An example is the antibody-antibiotic conjugate DSTA4637S that targets intracellular *S. aureus*, which can avoid eradication by current standard-of-care antimicrobials. This conjugate is constituted by a human IgG1 monoclonal antibody anti-*S. aureus* allied with a novel rifamycin-class antimicrobial through a protease cleavable linker. The mAb specifically attaches to the α-N-acetylglucosamine sugar residues of teichoic acid, a major component of *S. aureus* cell wall. When the phagocytic cells incorporate the *S. aureus* with the antibody conjugate attached, intracellular cathepsins cleave the linker, releasing the antimicrobial that kills the intracellular bacteria [66]. This antibody underwent randomized, double-blind, placebo-controlled, single-ascending-dose phase I trial analyzing safety, immunogenicity and pharmacokinetics in healthy volunteers. The participants received single intravenous doses of 5, 15, 50, 100, and 150 mg/kg of DSTA4637S, or placebo, and after 85 days, no serious or severe adverse events occurred [66]. A different example for *S. aureus* is the humanized monoclonal antibody Tefibazumab that targets the surface-expressed adhesion protein clumping factor A. This therapy is being developed as an adjunctive therapy for serious *S. aureus* infections. These mAbs were used in a phase II, randomized, double-blind clinical trial with the objective of *S. aureus* bacteremia treatment. The sixty patients enrolled in the study received a concentration of 20 mg/kg of body weight in a single infusion of either tefibazumab or a placebo in addition to an antibiotic. The main goal of the study was the determination of safety and pharmacokinetics. The study found no differences in adverse clinical events or alterations in laboratory values between the treatment groups, with four placebo patients showing progression in the severity of sepsis and none of the tefibazumab-treated patients exhibited this progression. Tefibazumab was found to have a safety profile comparable to other monoclonal antibodies. The authors suggest there is sufficient evidence to warrant further study in a larger trial to address the dosing range and efficacy [67]. 

A different approach using IgA antibodies was studied to confer protection against multidrug-resistant *Mycobacterium tuberculosis* infections. This strategy used the human monoclonal IgA 2E9 antibody targeting the alpha-crystallin (Acr, HspX) antigen in combination with the mouse interferon-gamma (IFN-*γ*). The studies were performed in mice transgenic for the human IgA receptor, CD89, and found the effect of the combined treatment was strongest when therapy was applied at the time of infection with reductions of 50-fold. Nonetheless, when therapy was initiated with an already established infection, a statistically significant reduction of lung bacterial load was observed [68].

## 5. Avian Immunoglobulin Y Antibodies

Immunoglobulin Y (IgY) is a isotype of immunoglobulin that can be found in birds. Antigen-specific IgY can be obtained from eggs laid by hens immunized with the selected antigens and be produced on a large scale. Three isotypes of immunoglobulins are present in these animals, being distinguishable in structure, immunochemical function, and concentration. These isotypes are IgA, IgM, and IgY, with the last one making up to 75% of the total immunoglobulin pool. The IgA and IgM found in birds are similar to mammalians in molecular weight and structure. The IgY was historically called IgG due to similarities in function and serum concentration. However, this is now considered incorrect, especially due to structural differences between the two (Figure 2) [69]. IgY (~180 kDa) is heavier than the mammalian IgG (~150 kDa). Structurally, IgY comprises two identical heavy chains and two identical light chains, linked by a disulfide bridge. Similar to the mammalian IgG, the IgY light chain is constituted by one variable domain and one constant domain. However, the intra-chain disulfide linkage between these two domains is absent in the IgY, leading to a more unstable structure with weaker intra-molecular forces. Unlike the mammalian IgG, the IgY heavy chain is composed of one variable domain and four constant domains, contrasting with the three constant domains found in IgG [70]. 

In 1893, Klemperer demonstrated that an immunized hen was able to transfer specific antibodies from the serum to the egg yolk [71], giving rise to the idea of the use of egg antibodies for therapies. The immunoglobins are unevenly distributed within the egg, IgA and IgM are incorporated in the egg white and the IgY in the egg yolk. The IgA (~0.7 mg/mL) and IgM (~0.15 mg/mL) have relatively low concentrations in the egg white, while the concentration of IgY in the egg yolk is significantly higher, ranging from 8 to 25 mg/mL [72]. Currently, the polyclonal antibodies available for usage in immunotherapies are mainly mammalian. The procedure for obtaining these molecules requires the performance of two steps in the IgG donor animal, which cause distress to the animal. This procedure includes immunization and the sacrifice or repeated bleeding. The adoption of polyclonal IgY for the development of antibody-based immunotherapies would lead to an increase of animal welfare, as the process for attaining the antibodies would be replaced by the collection of eggs. A reduction of the total number of animals used would also be accomplished, since the antibody productivity in this approach is 18 times superior to antibody production in rabbits [73]. The amount of antibodies present in the yolk is so high: 100 mg of antibodies can be obtained from a single egg, with a single hen producing around 20 eggs per month, making the production of IgY highly efficient [74]. The production of IgY starts by immunizing the hens with a target antigen (Figure 3). The immune response of the hens will be variable and dependent of several factors such as the antigen dose (with too much or not enough antigen inducing suppression, sensitization, or tolerance), the use of an adjuvant, the route of application (with the most common being injection of the breast muscle), the age and breed of the hen, and finally, immunization frequency and the interval between immunizations. The total number of required immunizations is variable but at least two immunizations are required. The interval between these immunizations is also important, being often reported to range from two to four weeks [75,76]. Booster immunization during the laying period can be performed to maintain the levels of specific antibodies up to a year [73]. 

The process for the isolation of IgY begins with the separation of lipids and granulate proteins of the egg yolk, obtaining the remaining water-soluble fraction. This fraction is a crude Ig concentrate. To obtain pure IgY, several methods can be performed. IgY can be separated from the water solute fraction by chromatography, filtration, or precipitation with PEG or salts such as ammonium or sodium sulphate [75]. The chosen method is highly dependent on the scale, quality, and cost effectiveness of the extraction [69,75,76]. A very important factor in the widespread use of antibody therapies is the stability of such antibodies. IgY has been studied in this regard and found to be stable during storage and processing. These antibodies maintain their activity after 6 months at room temperature or 1 month at 37 °C. At 4 °C, they can be stored up to a few years with the addition of 0.02% NaN_3_, 0.03% thimerosal, or gentamicin to prevent bacterial growth. Stability is not affected by freezing and freeze-drying if not extensively repeated. IgY are heat stable. However, at very high temperatures, the stability and binding activity decrease. The suggested method for long term storage is −20 °C, as lower temperatures of −70 °C caused loss of activity of around 50% [75,76,77,78]. 

As IgY are not mammalian, it is possible to obtain antibodies against highly conserved mammalian proteins or proteins able to evade the mammalian immune system. These antibodies are also capable of interacting with more epitopes on mammalian antigens, reducing the required amount of antibodies for an appropriate immune response [79]. Despite lacking recognition of mammalian Fc receptors, IgY do not prompt the mammalian complement activation, avoiding adverse inflammatory responses [79,80]. Recombinant humanized IgY are also being developed, where the constant domain of the IgY is substituted with the corresponding human domain. This allows the combination of the advantages of IgY antibodies with antibodies more suitable for in vivo therapeutics in humans. Similarly, monoclonal IgYs have been developed, having a higher affinity and specificity when compared with their polyclonal equivalents [81,82]. 

The development of therapeutics using IgY antibodies has been developed for a few years and has been successfully employed for prophylaxis and treatment of various enteric infections in animals such as cattle and swine. Using oral administration, specific IgY protects pigs and newborn calves from diarrhea caused by enterotoxigenic *E. coli* [75]. In humans, there have been IgY antibodies developed against multidrug-resistant bacteria, which reached clinical trials with different levels of success [79,83]. This is the case of IgY against *C. difficile* that is at phase II clinical trial (NCT04121169) using IgY polyclonal antibodies in increased doses administered twice a day for 10 to 14 days. Resolution of diarrhea, other symptoms, and fecal test parameters were used to assess clinical effectiveness. The results of this clinical trial are not available; however, previous studies had shown significant clinical improvement with no bacterial relapse [79]. 

A different double-blind, phase III clinical trial (NCT01455675) against *P. aeruginosa* was described, in which an IgY solution was gargled and swallowed every night for two minutes for a total of two years. The subjects were examined every 3 months regarding safety and efficacy. The results showed a good toleration profile for IgY against *P. aeruginosa*; however, lacked a clear demonstration of a therapeutic benefit in patients suffering from cystic fibrosis [79]. Less advanced studies have shown great promise. For example, in *A. baumannii*, specific IgY antibodies were produced by immunizing hens with formaldehyde inactivated bacteria, followed by purification from yolks using salt precipitation and ultracentrifugation. The antibodies were able to inhibit in vitro bacterial growth and significantly reduced mortality in BALB/c mice with an induced acute pneumonia caused by *A. baumannii* after intraperitoneal injection with specific IgYs [84]. Burned mice immunized with egg yolk specific antibodies raised against the *P. aeruginosa* OprF protein showed survival rates of 87.5% upon infection with the bacterium, compared to 25% for mice immunized with a control IgY [85].

## 6. Single-Domain Antibodies

Single-domain antibodies (sdAbs) or nanobodies are antibodies presenting only one monomeric variable domain, making them smaller than normal antibodies but retaining the ability to selectively bind to specific antigens. These immunoglobulins have evolved to attach to specific antigens using only three complementarity-determining region (CDR) loops, alternatively to the six present in conventional antibodies [86]. The existence of cryptic epitopes, consisting of narrow cavities (canyons) in the surface of antigens of several pathogens, able to bind to target receptors but inaccessible to intact antibodies, renders them generally immune-silent. These features have caused the interest of several researchers and led them to try to develop single domain antibodies. However, these fragments were mainly laboratory curiosities owing to their poor solubility, susceptibility to aggregation, and the fact that they rarely retained affinity [87]. This changed in 1993 when Raymond Hamers observed, in healthy dromedaries, a smaller isotype of IgG that lacked the light-chain and the first heavy-chain constant domain. This isotype comprised 75% of the total serum IgG. These antibodies, called HCAbs (heavy-chain only antibodies), were later found in other camelid species (llama and alpaca), albeit at lower concentrations (25% to 50%) [88]. The variable domain region of HCAbs presents unique sequences and was designated as VhH to distinguish them from the conventional VH domains. A few years after this discovery, a similar immunoglobulin was discovered as being part of the immune system of nurse sharks labeled IgNAR. These antibodies have a variable domain designated vNAR, followed by five constant domains, contrasting with the two constant domains of both HCAbs and normal antibodies, as depicted in Figure 2. Further studies revealed that vNAR have biophysical and chemical properties similar to VhH, such as high affinities and specificities, small size, and high thermal stability [89]. Both these variable domains have approximately 12 to 15 kDa, can be recombinantly produced, and are capable of recognizing antigens in the absence of the remainder of its heavy chain [86]. 

VhH are the smallest natural antigen binding entities, with a 2.5 nm diameter and 4 nm length [90]. Early structural studies of these fragments indicated that the interaction with antigens used mechanisms distinct from conventional antibodies. The specific mechanism used by sdAbs to bind to antigens is still not fully understood, and the most accepted general function is protein cleft recognition [91,92]. The paratopes of conventional antibodies against folded proteins present flat or concave structures, with convex binding sites being hard to achieve by murine and human antibodies. Synthetic antibodies can be engineered to present convex structures. Contrarily, sdAbs can adopt both flat and convex topologies, with concave being only inefficiently obtained [86,91,93]. In terms of amino acid content, no differences are observed between sdAbs and conventional antibodies, with the sdAbs paratopes having smaller molecular surface areas and smaller diameters. This difference leads to smaller footprints on the antigens targeted by sdAbs [86]. The binding between the antibodies and antigens is similar between the two immunoglobulin isotypes, using non-covalent interactions. These are present at a higher concentration in sdAbs due to the smaller paratopes, allowing for high-affinity interactions [86]. 

Notwithstanding, one of the major advantages of sdAbs is not how they bind to antigens but the accessibility to conserved cleft regions and pockets, such as binding sites and enzymes active sites, not available to traditional antibodies. This access is granted by their compact paratope diameter and long surface loops, usually larger than in conventional antibodies [94]. In addition, unlike mouse Vh domais, VhH and vNAR are stable, soluble, and easy to produce in vitro, rendering them as great resources for development of sensitive diagnostic platforms and sensors [94,95]. From a manufacturing perspective, sdAbs are inexpensive and simple to produce. The lack of post-translational modification allows their synthesis by a microbial system and the generation of a homogeneous product [96]. These antibodies have the combined advantages of mAbs therapeutics with the targeting potential of nanoscale delivery, as their sizes and structures allow the access to hidden or hard to reach epitopes, with their high modularity allowing the development of more complex constructs [86,94,95]. 

The modularity of sdAbs enables effective generation of bispecific or multispecific recombinant antibodies, which might include the fusion of sdAbs to form dimers, trimers, or tetramers. There is also the possibility to make sdAbs-mAb hybrid fusions [48]. It is important to notice some limiting factors, namely nanobodies are more prone to a rapid renal clearance, limiting their therapeutic lifetime. This occurs as their size is below renal filtration molecular mass cut-off, highly reducing their half-life within the organism [48,95]. The lack of a Fc region, besides having an impact on reducing half-live, also means these antibodies cannot directly initiate the Fc-mediated immune response. Due to the high similarity with the human VH domain, small size and low agglutination, these antibodies tend to present low immunogenicity [97]. Nonetheless, to reduce this immunogenicity, sdAbs can be humanized or fully human [98,99]. This process may, however, decrease their activity and solubility [100]. A different approach for the use of these antibodies is through engineered probiotic bacteria secreting or displaying the recombinant antibody fragment. As these bacteria can reside within the microbiota of the intestine, the antibody can be present and administered during long periods of time [101,102]. 

All the characteristics of these antibodies render them great candidates for use in microchip technologies for detection and diagnosis of infections or cancer. Nevertheless, their therapeutical potential is also very promising, as it offers new binding specificities, especially to target antigen binding sites inaccessible to conventional antibodies, such as enzyme active sites, G protein-coupled receptors, and viral surface canyons [94]. The first VhH-based therapy was approved in 2018 by the European Medicines Agency (EMA), called caplacizumab and is used for a rare blood-clotting disease [103]. 

The majority of sdAbs currently being developed target cancer cells or human viruses [104]. The few targeting bacterial infections target their toxins. This is the case for *C. difficile*, where the toxin pair A and B is targeted. Yang et al. [105] isolated TcdA- and TcdB-specific alpaca VhHs and constructed a tetravalent and bispecific tandem linked molecule of four VhH. The termini of this molecule targeted TcdA and the middle targets TcdB. This antibody was able to neutralize both toxins from clinical *C. difficile* isolates to protect mice from a lethal systemic challenge of a mixture of both toxins at an antibody concentration of 3.2 μg/kg and to reverse *C. difficile* infection in mice after a single injection. A different study by Andersen et al. [102] isolated four VhHs from lamas against TcdB RBD after immunization with the whole toxin. As monomers, three were able to neutralize the toxin effect on MA-104 cells. When combined in doublet or triplets, no additive effect was observed. The antibodies were then expressed on the surface of *Lactobacillus* and the previous three retained their effect and the fourth became neutralizing. *Lactobacillus* expressing the antibodies were then used in a prophylactic oral treatment of a hamster model of *C difficile* infection, and a combination of two strains showed a delayed death of the challenged hamsters [48,102].

## 7. Conclusions and Future Perspectives

Antimicrobials are a cornerstone of modern medicine that allow for tremendous increases in life expectancy and quality of humanity. However, resistance to antimicrobials is challenging this development and is becoming an ever-increasing problem that is estimated to surpass the combined deaths of cancer and heart disease by the year 2050, with an estimated 10 million deaths per year [22]. Every passing year, more antimicrobial agents are becoming ineffective. This problem is exacerbated by the ESKAPE group of bacteria, that, in some cases, are resistant to all clinically available antimicrobials [8]. This is a problem that requires the utmost attention, as well as new, innovative, and effective approaches, to be surpassed. Immunotherapies are seen by many as the optimal solution, and several approaches being studied are summarized in Table 1. 

Antibody characteristics allow for the targeting of only the bacteria of interest, not affecting the commensal bacteria of the host, while the development of resistance to antimicrobials is extremely unlikely. Antibody-based therapies have the advantage over vaccines of having an immediate response, allowing their use when the patient is already suffering from infection. 

One of the major drawbacks in the development and widespread use of antibody-based therapies is the upfront cost and research time required, combined with the high cost and difficulty of scaling up production [32]. Despite having a smaller cost of development when compared to vaccines, the upfront cost is still large. Nonetheless, the hospital stay and treatment offered to solve multidrug-resistant (MDR) infections is estimated to cost USD 55 billion (20 billion in health service cost and 35 billion in lost productivity) per year in the United States alone [109], adding some perspective on the economic problem that MDR is and will cause. Another problem raised against antibody-based therapies is the narrow spectrum they offer, only targeting one species of bacteria, requiring the knowledge of the agent causative of infection. However, the improvements in diagnostic methods, such as PCR or diagnostic antibodies, have proven that rapid diagnostic is possible and effective. The interplay between constant development and advances in research, and the many possibilities of antibody formats available with their advantages and disadvantages (Figure 4), will help to overcome the high cost and other obstacles required for the widespread use of these approaches. 

In the future, the MDR problem will not disappear, and without an upfront investment in the development of new immunotherapies and structures for their progress, humanity is risking going back to the medicine before the rise of antimicrobials. Notwithstanding, the growing clinical need, the increasing number of antibodies approved for therapeutic use, and the many technological advances in the field of immunoglobulin research, allows to envision a future with a widespread use of antibody-based therapies for bacterial infections.

## Figures and Tables

**Figure 1 vaccines-10-01789-f001:**
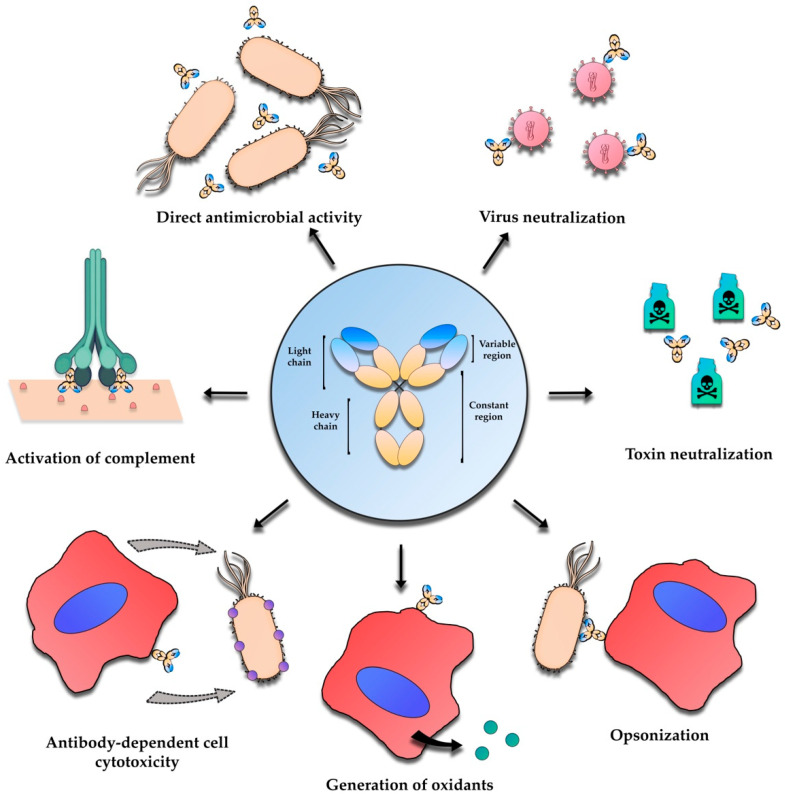
Biological roles played by antibodies. Opsonization and antibody-dependent cell cytotoxicity are dependent of mediators and host cells, while direct antimicrobial activity, generation of oxidants, complement activation, and virus and toxin neutralization are independent of other components of the immune system.

**Figure 2 vaccines-10-01789-f002:**
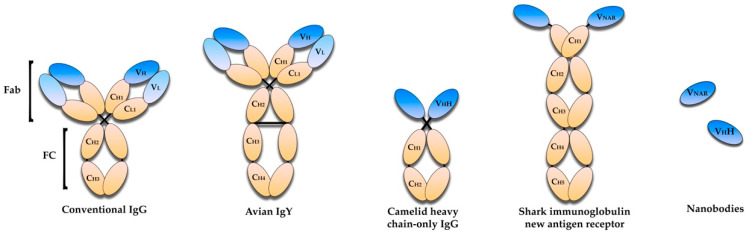
Domain structures of conventional mammalian tetrameric IgG, avian IgY, camelid heavy chain-only IgG (HCAb), shark immunoglobulin new antigen receptor (IgNAR), and nanobodies derived from HCAb and IgNAR. The variable domains are represented in blue and the constant domains are represented in yellow. The VHH and VNAR domains are stable and able to recognize antigens in the absence of the heavy chain.

**Figure 3 vaccines-10-01789-f003:**
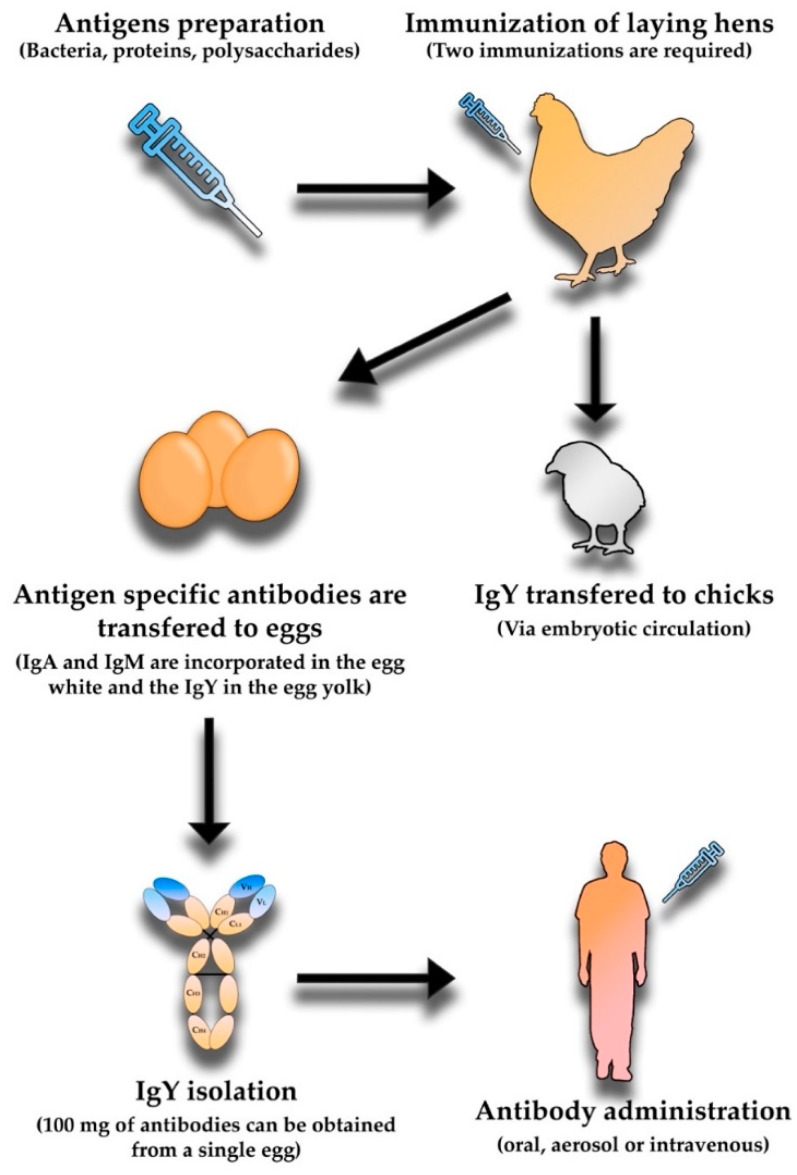
Summary of polyclonal IgY production process.

**Figure 4 vaccines-10-01789-f004:**
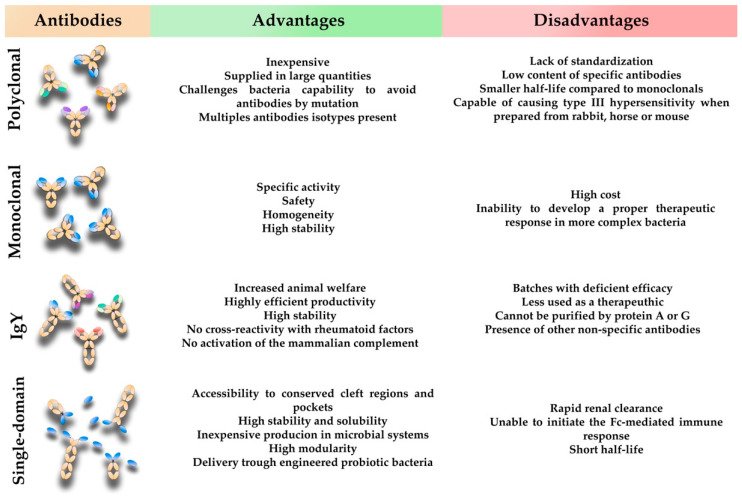
Main advantages and disadvantages of the four types of antibody formats discussed in this work.

**Table 1 vaccines-10-01789-t001:** Passive immunization preclinical and clinical trials mentioned in the work.

Bacteria	Antibody Format	Immunogen	Clinical Trial	Results	References
**Gram-positive bacteria**					
*C. difficile*	Polyclonal	Anti-toxin B goat antibodies	Preclinical	Hamsters challenged with toxin B showed survival rates of 98%	[106]
	Monoclonal	Bexlotoxumab (Enterotoxin B)	FDA approved	Approved for reduction of recurrence of infection	[52]
	IgY	IM-01 (Toxin A and B and spore preparation)	Phase II (NCT04121169) Randomized, parallel assignment, open-label trial	Clinical improvement and no relapse of infection	[79]
	Single-domain antibodies	Tetravalent and bispecific tandem linked molecule of four VhH against toxin A and B	Preclinical	Protect mice from a lethal systemic challenge of a mixture of both toxins and reverse infection in mice	[105]
	Single-domain antibodies	4 VhHs from lamas against toxin B expressed on the surface of *Lactobacillus*	Preclinical	Delayed death of the hamsters challenged	[102]
*S. aureus*	Polyclonal	Altastaph (capsular polysaccharides)	Phase II (NCT00063089) double-blind, placebo-controlled trial	Not powered to show efficacy, safety profile suggests that Altastaph may be an effective adjunct to antibiotics	[42]
	Polyclonal	Veronate^®^ (surface adhesin)	Phase III trial (NCT00113191) double-blind, comparing the safety and efficacy versus placebo	Exhibited no effect in reduction of *S. aureus* prevention of late-onset sepsis in very low birth weight infants	[43]
	Monoclonal	DSTA4637S (human anti-*S. aureus IgG1* allied with a novel rifamycin-class antibiotic)	Phase I (NCT02596399) randomized, double- blind, placebo-controlled, single-ascending-dose	Safety and pharmacokinetic profile favorable for development of new therapeutic	[66]
	Monoclonal	Tefibazumab (surface-expressed adhesion protein clumping factor A)	PhaseII (NCT00198302) randomized, double-blind, placebo-controlled clinical trial with the objective of *S. aureus* bacteremia treatment.	Well tolerated, with safety profile similar to other monoclonal antibodies. Further trials are necessary for dose range and efficacy	[67]
*B. anthracis*	Monoclonal	Raxibacumab (Toxin)	FDA approved	Approved for treatment of anthrax inhalation as a result of *B. anthraci.*	[50]
	Monoclonal	Obiltoxaximab (Toxin)	FDA approved	Approved for treatment of anthrax inhalation as a result of *B. anthraci.*	[51]
*M. tuberculosis*	Monoclonal	Human monoclonal IgA 2E9 and Interferon-γ	Preclinical	Reduction of 50-fold of lung bacterial load when applied at the time of infection	[68]
**Gram-negative bacteria**					
*E. coli*	Polyclonal	Hyperimmune anti-Stx2 bovine colostrum	Preclinical	Prevention of 100% of the lethality caused by *E. coli* O157:H7 in a weaned mice model	[107,108]
*B. cenocepacia*	Polyclonal	Goat anti-OmpA-like protein	Preclinical	In vitro greatly impairs the ability to adhere and invade human epithelial cells	[41]
*P. aeruginosa*	Monoclonal	MAb 166 (Murine monoclonal antibody to PcrV)	Preclinical	Protective when intraperitoneally transferred to mice	[59]
	Monoclonal	KB001-A (anti-PcrV PEGylated mouse Mab	Phase II (NCT01695343) double-blind, placebo-controlled trial	Prevents ventilator-associated pneumonia, the efficacy was low in patients suffering from CF	[53,60]
	Monoclonal	Panobacumab (IgM targeting the O-antigen of serotype O11)	Phase II (NCT00851435) safety and PK in patients with hospital acquired pneumonia	Improve clinical outcome in a shorter time.	[65]
	IgY	PsAer-IgY (anti-pseudomonas antibodies)	Phase III (NCT01455675) randomized, parallel assignment, double-blind trial	Good toleration profile, lacked a clear demonstration of a therapeutic benefit in CF patients	[79]
	IgY	Anti-OprF antibodies	Preclinical	Increased survival rates, in burned mice infected with the bacteria	[85]
*A. baumannii*	IgY	Specific anti-*A. baumannii* antibodies	Preclinical	Reduced mortality in BALB/c mice with an induced acute pneumonia after intraperitoneal injection with specific IgYs	[84]

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
