# Peer review of "Antibody-Based Immunotherapies as a Tool for Tackling Multidrug-Resistant Bacterial Infections"

_vaccines, 2022, doi:10.3390/vaccines10111789_

Round 1

Reviewer 1 Report

The review by Seixas A.M.M. et al entitled “Antibody-based immunotherapies as a tool for tackling multi-2 drug-resistance bacterial infections” proposes to discuss and summarize the information on antibody-based immunotherapies. The topic is interesting since vaccines are the most successful treatment for infections with the advantage of generating immunological memory. Several previous reviews cover a similar issue (McCulloch TR et al 2021 Trends in Microbiology; Wallis RS et al 2022 Nature Reviews Immunology), however, the current manuscript can complement updated information about clinical trials and present a suitable didactic text for the general public.

For the topic researchers, the text doesn´t present depth knowledge of novelties, it is more of an overview of the antibody-based immunotherapy context.

The text is fairly written and the figures are appropriate and easy to understand.

Author Response

REVIEWER #1

Reviewer: The review by Seixas A.M.M. et al entitled “Antibody-based immunotherapies as a tool for tackling multi-2 drug-resistance bacterial infections” proposes to discuss and summarize the information on antibody-based immunotherapies. The topic is interesting since vaccines are the most successful treatment for infections with the advantage of generating immunological memory. Several previous reviews cover a similar issue (McCulloch TR et al 2021 Trends in Microbiology; Wallis RS et al 2022 Nature Reviews Immunology), however, the current manuscript can complement updated information about clinical trials and present a suitable didactic text for the general public. For the topic researchers, the text doesn´t present depth knowledge of novelties, it is more of an overview of the antibody-based immunotherapy context. The text is fairly written and the figures are appropriate and easy to understand.

Response: We kindly appreciate the general comments made by the reviewer. Indeed, we are aware of nice revisions on the subject and we wanted to submit a manuscript that goes further in updated information, while keeping its didactic interest for the general public.

Reviewer 2 Report

General comment: 

The lack of paragraphing makes reading the manuscript tiresome. It does not provide a clear direction and instructive model for readers to follow the trail of thoughts by the authors. Restructuring of the sections and use of paragraphing would be very helpful for readers to follow the authors intended vision of the review.

In the introduction, the mention of phage therapy as another method for therapy should be included to ensure an all encompassing review.

Line 90. The authors only focused on IgG and IgM but IgA is also another important isotope that would be beneficial for therapy especially it works on mucosal immunity against bacterial infection.

Paragraphing to introduce the mechanisms would help readers. 

Line 153: The half lives is not entirely accurate: In some cases half lives are only days depending on format. Human igg1 from 6 days to over a month (https://doi.org/10.1016/j.isci.2022.103746) whereas VHH only hours (Antibodies 2019, 8(1), 1; https://doi.org/10.3390/antib8010001)

Line 162: "Fortunately, the development of PCR and other rapid diagnostic techniques provide fast platforms helping the establishment of antibody-based therapies" How does diagnostics help in establishing antibody therapy?

Line 184: "antibody for passive immunotherapy requires a noticeably smaller time and cost than that needed to develop a vaccine" - this is partially accurate as antibiotics was not compared which is the mainstream treatment now. 

Line 187: The term antibody classes is loosely used where it was used to defined isotypes earlier and now is to define different formats or species derived antibodies. Please use correct terminology.

Fig 1: the manner in which Fab is labeled can be misleading to reflect only the two blue regions. identify the fab format clearer in figure.

Line 195: Pls rephrase sentence.

There is a concern here on based on the description of the sections. The Polyclonal antibody section is mainly referring to serum therapy which could be classed as Immune IVIG. However, the authors did not discuss the use of standard IVIG therapy for bacterimia espeically for patients with primary immune deficiencies. This section is rather messy where serum therapy by animals is mixed with human IVIG and categorised as polyclonal antibody all the same. 

Line 256: The technology for monoclonal antibodies (mAbs) production was introduced in 1975 by Kohler and Milstein and consisted in a mouse hybridoma. What do you mean by this?

The monoclonal antibody section is hard to follow. Structure the thoughts. Classify the mabs properly. Mabs can be for IgY and single domain. IgY can also be used as polyclonal. So to having it as an independent section is misleading. 

IgY is also mainly polyclonal in nature. It can only be selected as monoclonal when recombinant methods are used like phage display and other display methods. 

Single domain antibodies can also be human based and not just camelid or shark antibodies.

There is a lot of information that the authors are trying to put together but the classification of antibodies in terms of polyclonal, monoclonal, igy and domain antibodies is misleading.

I am not sure what the focus of the review is. Is it on different antibody structures, development methods or poly- and mono-specificity antibodies. I would also like to highlight that the use of paragraphing would greatly help readers to follow the topics in a structured manner.

Author Response

REVIEWER #2

Reviewer: General comment: 

The lack of paragraphing makes reading the manuscript tiresome. It does not provide a clear direction and instructive model for readers to follow the trail of thoughts by the authors. Restructuring of the sections and use of paragraphing would be very helpful for readers to follow the authors intended vision of the review. In the introduction, the mention of phage therapy as another method for therapy should be included to ensure an all encompassing review.

Response: Thanks for the general appreciation of the work and identification of some weak points. We have prepared the revised version taking these comments in consideration. We agree that the addition of phage therapy, an interesting alternative against antibiotic-resistant bacterial infections, would broaden the review scope. However, the present review is focused on Antibody-based immunotherapies, which does not include phage therapies, another antibiotic alternative therapy to treat MDR bacterial infections. We will consider this suggestion in a future revision concerning alternative methods to tackle MDR bacteria.

Reviewer: Line 90. The authors only focused on IgG and IgM but IgA is also another important isotope that would be beneficial for therapy especially it works on mucosal immunity against bacterial infection.

 Response: Thanks for the suggestion. Taking it into consideration, new data about IgA-based immunotherapies were added to the section 2: “Antibody based immunotherapies - an overview” and to section 4: “ Conventional Mammalian Monoclonal Antibodies”. See new lines 114 to 135 and 648 to 655.

Reviewer: Paragraphing to introduce the mechanisms would help readers.

Response: Thanks for the suggestion. We have introduced additional paragraphing throughout the text in the revised version of the manuscript. 

Reviewer: Line 153: The half lives is not entirely accurate: In some cases half-lives are only days depending on format. Human igg1 from 6 days to over a month (https://doi.org/10.1016/j.isci.2022.103746) whereas VHH only hours (Antibodies 2019, 8(1), 1; https://doi.org/10.3390/antib8010001)

Response: Thanks for the excellent observation. The text was modified and now reads as: “Despite some exceptions, such as some nanobodies, antibodies are significantly more stable than antimicrobials, having half-lives from weeks to months, exceeding those of the antimicrobials that only maintain their activity in the patient for a few days.” See new lines 237 and 238.

Reviewer: Line 162: "Fortunately, the development of PCR and other rapid diagnostic techniques provide fast platforms helping the establishment of antibody-based therapies." How does diagnostics help in establishing antibody therapy?

Response: Thanks for the question. We added the following sentence to explain how these methods contribute to the establishment of antibody-based therapies: “Indeed, these fast diagnostic methods allow the specific identification of the pathogen in a short time, enabling the accurate selection of a specific anti-body-based therapy. Altogether, this combination increases the confidence and reliability of anti-body-based therapies therapies, thus contributing to their establishment.” See new lines 249-253.

Reviewer: Line 184: "antibody for passive immunotherapy requires a noticeably smaller time and cost than that needed to develop a vaccine" - this is partially accurate as antibiotics was not compared which is the mainstream treatment now. 

Response: Thanks for the observation, which is correct. However, in this sentence we are comparing the development of vaccines with antibody-based immunotherapies, and not with antibiotics.

Reviewer: Line 187: The term antibody classes is loosely used where it was used to defined isotypes earlier and now is to define different formats or species derived antibodies. Please use correct terminology.

Response: Thanks for the suggestion, which we have taken into consideration in the revised manuscript.

Reviewer: Fig 1: the manner in which Fab is labeled can be misleading to reflect only the two blue regions. identify the fab format clearer in figure.

Response: Thanks for the comment. In our view, in Fig. 1 the fab region seems clearly identified as the bracket is in both the blue and yellow domains. Please note that the blue regions are described as the variable domains in the figure legend.

Reviewer: Line 195: Pls rephrase sentence.

Response: Thanks for the suggestion. The sentence was re-written.

Reviewer: There is a concern here on based on the description of the sections. The Polyclonal antibody section is mainly referring to serum therapy which could be classed as Immune IVIG. However, the authors did not discuss the use of standard IVIG therapy for bacterimia espeically for patients with primary immune deficiencies. This section is rather messy where serum therapy by animals is mixed with human IVIG and categorised as polyclonal antibody all the same. 

 Response: Thanks for the observation. We started the section by mentioning that pAbs can be purified from animal sera. We have not mentioned the direct administration of sera from animals, but the purified animals. We have shorten the section, as our purpose was not the discussion of IVIG therapies, but the use of purified antibodies.

Reviewer: Line 256: The technology for monoclonal antibodies (mAbs) production was introduced in 1975 by Kohler and Milstein and consisted in a mouse hybridoma. What do you mean by this?

Response: Thanks for the remark. The sentence was re-written and now reads: “: The technology for monoclonal antibodies (mAbs) production was introduced in 1975 by Kohler and Milstein”.

Reviewer: The monoclonal antibody section is hard to follow. Structure the thoughts. Classify the mabs properly. Mabs can be for IgY and single domain. IgY can also be used as polyclonal. So to having it as an independent section is misleading. 

Response: We kindly appreciate the comment. Therefore, the manuscript was restructured in Conventional mammalian polyclonal antibodies, Conventional mammalian monoclonal antibodies, IgY and single-domain antibodies. The IgY and single-domain antibodies have different characteristics from the other conventional mammalian antibody formats mentioned, so they were added to independent sections.

Reviewer: IgY is also mainly polyclonal in nature. It can only be selected as monoclonal when recombinant methods are used like phage display and other display methods. 

Response: Thanks for the comment. In the text this is mentioned in lines 690, 694, and 843.

Reviewer: Single domain antibodies can also be human based and not just camelid or shark antibodies.

Response: Thanks for the comment This is mentioned in the text lines 914-915.

Reviewer: There is a lot of information that the authors are trying to put together but the classification of antibodies in terms of polyclonal, monoclonal, igy and domain antibodies is misleading. I am not sure what the focus of the review is. Is it on different antibody structures, development methods or poly- and mono-specificity antibodies. I would also like to highlight that the use of paragraphing would greatly help readers to follow the topics in a structured manner.

Response: Thanks for the comments. In order to accommodate these criticisms, the manuscript was restructured in Conventional mammalian polyclonal antibodies, Conventional mammalian monoclonal antibodies, IgY and Single-domain antibodies, and paragraphing was used throughout the text.

Reviewer 3 Report

Nice and balanced review.

There are small grammatical issues and Fig. 2 needs to be changed a little (see enclosed pdf, which can be forwarded to thre authors).

Abbreviations must be defined at first mention.

Author Response

REVIEWER # 3

Reviewer: Nice and balanced review. There are small grammatical issues and Fig. 2 needs to be changed a little (see enclosed pdf, which can be forwarded to the authors).

Response: The authors kindly appreciate the comments of the reviewer. In figure 2, we have included nanobody as a separated group .  VhH and VnAR are represented in the image and we have also slightly modified the legend, which now reads as follows: “Domain structures of conventional mammalian tetrameric IgG, avian IgY, camelid heavy chain-only IgG (HCAb), shark immunoglobulin new antigen receptor (IgNAR) and nanobodies derived from HCAb and IgNAR. The variable domains are represented in blue and the constant domains are represented in yellow. The VHH and VNAR domains are stable and able of recognizing antigens in the absence of the heavy chain.”

Reviewer: Abbreviations must be defined at first mention.

Response: Thanks for the observation. We have revised the manuscript in order to define each abbreviation the first time it appears.

Round 2

Reviewer 2 Report

The revision has substantially improved the flow of the manuscript but there are still some sections that require some changes. In it current form, the manuscript is still not ready for publication.

Section 3 would read better and less confusing by separating the animal derived polyclonal antibodies via immunisation and from humans either via convalescent patient sera, IVIG from healthy humans or even vaccinated individuals. Suggest to use paragraphing to separate the different approaches.

Section 4 should be Human Monoclonal antibodies- there is no mention on the how these chimeric, humanised or human antibodies are made.  

Section 5 can be avian IgY antibodies

Author Response

POINT-BY-POINT RESPONSE TO REVIEWER #2 CRITICISMS

Reviewer: The revision has substantially improved the flow of the manuscript but there are still some sections that require some changes. In it current form, the manuscript is still not ready for publication.

Section 3 would read better and less confusing by separating the animal derived polyclonal antibodies via immunisation and from humans either via convalescent patient sera, IVIG from healthy humans or even vaccinated individuals. Suggest to use paragraphing to separate the different approaches.

Response: The authors gratefully acknowledge the suggestions. We have made substantial changes in the sequence of sentences. The whole section, spanning lines 232 to 298 now reads as follows“The first antibody preparations used for passive immunization contained polyclonal antibodies derived from the sera of immunized animals or humans, and in some cases convalescing patients [2]. Nowadays, pAbs are produced by injecting an immunogen into an animal to elicit an immune response against a specific antigen. After immunization, the polyclonal antibodies are purified to obtain a solution that is free from other serum proteins. These antibodies can be purified directly from an animal serum after immunization with the antigen of interest or the infectious agent. The term polyclonal is derived from the fact that these molecules are derived from multiple B cell clones and can recognize different epitopes from the same antigen. This occurs because different lymphocyte clones responding against different epitopes of the antigen are formed during the antibody response [11,34].

pAbs preparations are comprised of numerous antibodies differing in primary structure, isotype, specificity, and glycosylation of the constant region, an important factor for interaction with Fc receptors [35]. The protective effect of pAbs is observed naturally in the passive transference of maternal immunity to protect newborns during the more vulnerable phases of early life [36]. The preparation of pAbs for clinical uses starts with the immunization of an animal. As higher volumes of blood can be retrieved from larger animals, the traditional sources of these antibodies are horses, sheep, goats, and rabbits, rather than mice [34]. This production is relatively inexpensive, and the total amount produced is limited by the amount of blood removed from the animal during its lifetime. Once an animal is exhausted, new immunization in a different animal needs to be performed which will not be identical to the previous, due to the stochastic nature of the adaptive immune response and the polyclonal nature of these antibody preparations [34]. The method through which pAbs are produced makes them cost-effective, able to be supplied in large quantities and the usage of larger animals reduces the need for multiple batches to be validated. Presently, pAbs are a low-cost alternative to monoclonal antibodies which are usually used for snake bites and post-exposure prophylaxis of infectious diseases like rabies, botulism, and diphtheria [37].

The ability of these antibodies to recognize several epitopes also challenges the pathogen capability to avoid antibodies by mutating the antigen [38]. The variability inherently present in pAbs preparations increases the biological effector functions, as multiples subclasses of IgG antibodies are present (IgG1, IgG2, IgG3 and IgG4) providing in many cases added benefits and protective advantage [39]. The use of these pAbs is also associated with some limitations, namely the lack of standardization due to the lot-to-lot variation could lead to deficient efficacy and low content of specific antibodies.

The adoption of approaches using pAbs has been greater for tackling virus like influenza, HIV, MERS and SARS, with some levels of success [19,35,36,39,40], but also bacterial toxins like shiga toxin and toxins from Clostridium difficile [11,19]. The use for treatment of other bacterial infections has been less common. Our group recently showed that a pAbs against an OmpA-like protein from Burkholderia cenocepacia greatly impairs the ability of these bacteria to adhere and invade human epithelial cells in vitro [41]. Both the adhesion and invasion are highly important steps in the infection process.

The fact that the antibodies produced, usually, are non-human may lead to exacerbated immune responses in the patients. This last hurdle could be overcome with the production of humanized antibodies in the animals. Beigel et al used a novel approach, with transchromosomic cattle carrying a human artificial chromosome comprising the entire human immunoglobulin gene repertoire, human immunoglobulin heavy chain (IGH) locus from chromosome 14, and human k light chain (IGK) locus from chromosome 2.6 [40]. These authors reported the production of a highly potent and specific fully human polyclonal IgG using this system [40].

An example is the human polyclonal immune globulin preparation (Altastaph) that targets capsular polysaccharides of S. aureus and is being developed for S. aureus infections complicated by bacteremia. A phase II double-blind, placebo-controlled trial found that when compared to the placebo the patients had a shorter median time to the resolution of fever and a shorter length of hospital stay. Unfortunately, the study was not powered to show efficacy. Nonetheless, the preliminary findings and safety profile indicate the potential of this therapy as an adjunct to antimicrobials [42]. A different human polyclonal preparation named Veronate® against S. aureus derived from donors with high titers of antibodies against a surface adhesin of S. aureus and Staphylococcus epidermidis also went into clinical trials. The phase II trial showed promising results for the highest dose used. However, a phase III trial (NCT00113191) evaluating the prevention of late-onset sepsis in very low birth weight infants, showed no effect of the antibody treatment in reduction of S. aureus [43]. A possible explanation for the disappointing result is the requirement of a humoral, cellular, and phagocytic responses for control of these infections that is not properly provided by the pre-mature infants [19]. “

Reviewer: Section 4 should be Human Monoclonal antibodies- there is no mention on the how these chimeric, humanised or human antibodies are made.  

Response: Thanks for the suggestions. However,we decided to maintain section 4 title, since chimeric and humanized antibodies have a percentage of human sequences always below 100%. As such, they are not viewed in our perspective as fully human monoclonal antibodies.

In order to respond to the lack of information on how these chimeric, humanised or human antibodies are made, we included in new lines 366 to 380  short descriptions of the production of chimeric, humanised or human antibodies, which now reads as follows: “Chimeric antibodies are a combination of the murine variable domains fused to the human constant domains. These type of antibodies are produced by cloning the murine variable region heavy and light chains genes, amplified from the hybridoma, together with the constant region genes of human heavy and light chains into a plasmid. This plasmid is then transfected into bacteria where the antibodies are produced as inclusion bodies, and then purified for in vivo use. These molecules are around 70% human and possess a human Fc sequence, reducing the possibility of immunogenicity [46]. Humanized antibodies contain 85 to 90% of human sequences. Their production is achieved by replacing all the rodent sequences except the complementarity determining regions (CDRs) for human sequences. However, this process of humanization is technically challenging, the insertion of CDRs in a generic framework is insufficient, as antibody affinity sometimes relies on framework regions. This process leads to antibody activity losses. Conservation of a few rodent sequences in the framework is required to restore the binding [46-47]. The production of fully human monoclonal antibodies was achieved with the development of phage display platforms, and also with the advancements in transgenic mouse platforms [46].”

Reviewer: Section 5 can be avian IgY antibodies

Response: Thanks for the suggestion, we have performed the suggested change.